# A Convergent Single-Loop Algorithm for Relaxation of Gromov-Wasserstein in Graph Data

**Jiajin Li**
Stanford University
`jiajinli@stanford.edu`

**Jianheng Tang**
HKUST (GZ)
`jtangbf@connect.ust.hk`

**Lemin Kong**
CUHK
`lkong@se.cuhk.edu.hk`

**Huikang Liu**
SUFE
`liuhuikang@sufe.edu.cn`

**Jia Li**
HKUST (GZ)
`jialee@ust.hk`

**Anthony Man-Cho So**
CUHK
`manchoso@se.cuhk.edu.hk`

**Jose Blanchet**
Stanford University
`jose.blanchet@stanford.edu`

## Abstract

In this work, we present the Bregman Alternating Projected Gradient (BAPG) method, a single-loop algorithm that offers an approximate solution to the Gromov-Wasserstein (GW) distance. We introduce a novel relaxation technique that balances accuracy and computational efficiency, albeit with some compromises in the feasibility of the coupling map. Our analysis is based on the observation that the GW problem satisfies the Luo-Tseng error bound condition, which relates to estimating the distance of a point to the critical point set of the GW problem based on the optimality residual. This observation allows us to provide an approximation bound for the distance between the fixed-point set of BAPG and the critical point set of GW. Moreover, under a mild technical assumption, we can show that BAPG converges to its fixed point set. The effectiveness of BAPG has been validated through comprehensive numerical experiments in graph alignment and partition tasks, where it outperforms existing methods in terms of both solution quality and wall-clock time.

## 1 Introduction

The GW distance provides a flexible way to compare and couple probability distributions supported on different metric spaces. This has led to a surge in literature that applies the GW distance to various structural data analysis tasks, including 2D/3D shape matching (Peyré et al., 2016; Mémoli & Sapiro, 2004; Mémoli, 2009), molecule analysis (Vayer et al., 2018; 2019a), graph alignment and partition (Chowdhury & Mémoli, 2019; Xu et al., 2019b;a; Chowdhury & Needham, 2021; Gao et al., 2021), graph embedding and classification (Vincent-Cuaz et al., 2021b; Xu et al., 2022), generative modeling (Bunne et al., 2019; Xu et al., 2021).

Although the GW distance has gained a lot of attention in the machine learning and data science communities, most existing algorithms for computing the GW distance are double-loop algorithms that require another iterative algorithm as a subroutine, making them not ideal for practical use. Recently, an entropy-regularized iterative sinkhorn projection algorithm called eBPG was proposed by Solomon et al. (2016), which has been proven to converge under the Kurdyka-Łojasiewicz framework. However, eBPG has several limitations. Firstly, it addresses an entropic-regularized GW objective, whose regularization parameter has a major impact on the model's performance. Secondly, it requires solving an entropic optimal transport problem at each iteration, which is both computationally expensive and not practical. In an effort to solve the GW problem directly, Xu et al. (2019b) proposed the Bregman projected gradient (BPG), which is still a double-loop algorithm that relies on another iterative algorithm as a subroutine. Additionally, it suffers from numerical instability due to the lack of an entropic regularizer. While Vayer et al. (2019a); Mémoli (2011) introduced the

Frank-Wolfe method to solve the GW problem, they still relied on linear programming solvers and line-search schemes, making it unsuitable for even medium-sized tasks. Recently, Xu et al. (2019b) developed a simple heuristic, single-loop method called BPG-S based on BPG that showed good empirical performance on node correspondence tasks. However, its performance in the presence of noise is unknown due to the lack of theoretical support.

The main challenge lies in efficiently tackling the Birkhoff polytope constraints (i.e., the polytope of doubly stochastic matrices) for the coupling matrix. The key issue is that there is no closed update for its Bregman projection, which forces current algorithms to rely on computationally expensive or hyperparameter-sensitive iterative methods. To address this difficulty, we propose a single-loop algorithm (BAPG) that solves the GW distance approximately. Our solution incorporates a novel relaxation technique that sacrifices some feasibility of the coupling map to achieve computational efficiency. This violation is acceptable for certain learning tasks, such as graph alignment and partition, where the quality of the coupling is not the primary concern. We find that BAPG can obtain desirable performance on some graph learning tasks as the performance measure for those tasks is the matching accuracy instead of the sharpness of the probabilistic correspondence. In conclusion, BAPG offers a way to sacrifice the feasibility for both computational efficiency and matching accuracy.

In our approach, we decouple the Birkhoff polytope constraint into separate simplex constraints for the rows and columns. The projected gradient descent is then performed on a constructed penalty function using an alternating fashion. By utilizing the closed-form Bregman projection of the simplex constraint with relative entropy as the base function, BAPG only requires matrix-vector/matrix-matrix multiplications and element-wise matrix operations at each iteration, making it a computationally efficient algorithm. Thus, BAPG has several convenient properties such as compatibility with GPU implementation, robustness with regards to the step size (the only hyperparameter), and low memory requirements.

Next, we investigate the approximation bound and convergence behavior of BAPG. We surprisingly discover that the GW problem satisfies the Luo-Tseng error bound condition (Luo & Tseng, 1992). This fact allows us to bound the distance between the fixed-point set of BAPG and the critical point set of the GW problem, which is a notable departure from the usual approach of utilizing the Luo-Tseng error bound condition in establishing the linear convergence rate for structured convex problems (Zhou & So, 2017). With this finding, we are able to quantify the approximation bound for the fixed-point set of BAPG explicitly. Moreover, we establish the subsequence convergence result when the accumulative asymmetric error of the Bregman distance is bounded.

Lastly, we present extensive experimental results to validate the effectiveness of BAPG for graph alignment and graph partition. Our results demonstrate that BAPG outperforms other heuristic single-loop and theoretically sound double-loop methods in terms of both computational efficiency and matching accuracy. We also conduct a sensitivity analysis of BAPG and demonstrate the benefits of its GPU acceleration through experiments on both synthetic and real-world datasets. All theoretical insights and results have been well-corroborated in the experiments.

## 2 PROPOSED ALGORITHM

In this section, we begin by presenting the GW distance as a nonconvex quadratic problem with Birkhoff polytope constraints. We then delve into the theoretical insights and computational characteristics of our proposed algorithm, BAPG.

The Gromov-Wasserstein distance was first introduced in (Mémoli, 2011; 2014; Peyré et al., 2019) as a way to quantify the distance between two probability measures supported on different metric spaces. More precisely:

**Definition 2.1** (GW distance). *Suppose that we are given two unregistered compact metric spaces* $(\mathcal{X}, d_X)$, $(\mathcal{Y}, d_Y)$ *accompanied with Borel probability measures* $\mu, \nu$ *respectively. The GW distance between* $\mu$ *and* $\nu$ *is defined as*

$$\inf_{\pi \in \Pi(\mu, \nu)} \iint |d_X(x, x') - d_Y(y, y')|^2 d\pi(x, y) d\pi(x', y'),$$

*where* $\Pi(\mu, \nu)$ *is the set of all probability measures on* $\mathcal{X} \times \mathcal{Y}$ *with* $\mu$ *and* $\nu$ *as marginals.*

Intuitively, the GW distance aims to preserve the isometric structure between two probability measures through optimal transport. If there is a map that pairs $x \to y$ and $x' \to y'$, then the distance between $x$ and $x'$ should be similar to the distance between $y$ and $y'$. Due to these desirable properties, the GW distance is a powerful tool in structural data analysis, particularly in graph learning. Some examples of its applications include (Vayer et al., 2019b; Xu et al., 2019b;a; Solomon et al., 2016; Peyré et al., 2016) and related references.

To start with our algorithmic developments, we consider the discrete case for simplicity and practicality, where $\mu$ and $\nu$ are two empirical distributions, i.e., $\mu = \sum_{i=1}^{n} \mu_i \delta_{x_i}$ and $\nu = \sum_{j=1}^{m} \nu_j \delta_{y_j}$. As a result, the GW distance can be reformulated as follows:

$$\min_{\pi \in \mathbb{R}^{n \times m}} -\mathrm{Tr}(D_X \pi D_Y \pi^T)$$
$$\text{s.t.} \quad \pi 1_m = \mu, \ \pi^T 1_n = \nu, \ \pi \geq 0, \tag{1}$$

where $D_X$ and $D_Y$ are two symmetric distance matrices.

## 2.1 RELAXATION OF GW DISTANCE

Now, we will introduce our relaxation of GW distance. The nonconvex quadratic program (1) with polytope constraints is typically addressed by (Bregman) projected gradient descent type algorithms. However, existing algorithms require an inner iterative algorithm, such as Sinkhorn (Cuturi, 2013) or the semi-smooth Newton method (Cuturi, 2013), to solve the regularized optimal transport problem at each iteration. This can lead to a computationally intensive double-loop scheme, which is not ideal for GPU-friendly computation. To overcome this issue, we aim to handle the row and column constraints separately using an operator splitting-based relaxation technique.

For simplicity, we consider the compact form for better exploiting the problem specific structures:

$$\min_{\pi} f(\pi) + g_1(\pi) + g_2(\pi). \tag{2}$$

Here, $f(\pi) = -\mathrm{Tr}(D_X \pi D_Y \pi^T)$ is a nonconvex quadratic function; $g_1(\pi) = \mathbb{I}_{\{\pi \in C_1\}}$ and $g_2(\pi) = \mathbb{I}_{\{\pi \in C_2\}}$ are two indicator functions over closed convex polyhedral sets. Here, $C_1 = \{\pi \geq 0 : \pi 1_m = \mu\}$ and $C_2 = \{\pi \geq 0 : \pi^T 1_n = \nu\}$. To decouple the Birkhoff polytope constraint, we adopt the operator splitting strategy to reformulate (2) as

$$\min_{\pi = w} f(\pi, w) + g_1(\pi) + g_2(w) \tag{3}$$

where $f(\pi, w) = -\mathrm{Tr}(D_X \pi D_Y w^T)$. Then, we penalize the equality constraint and process the alternating minimization scheme on the constructed penalized function, i.e.,

$$F_\rho(\pi, w) = f(\pi, w) + g_1(\pi) + g_2(w) + \rho D_h(\pi, w).$$

Here, $D_h(\cdot, \cdot)$ is the so-called Bregman divergence, i.e., $D_h(x, y) := h(x) - h(y) - \langle \nabla h(y), x - y \rangle$, where $h(\cdot)$ is the Legendre function, e.g., $\frac{1}{2}\|x\|^2$, relative entropy $x \log x$, etc.

## 2.2 BREGMAN ALTERNATING PROJECTED GRADIENT (BAPG)

Next, we present the proposed single-loop Bregman alternating projected gradient (BAPG) method. The crux of BPAG is to take the alternating projected gradient descent step between $C_1$ and $C_2$. For the $k$-th iteration, the BAPG update takes the form

$$\pi^{k+1} = \underset{\pi \in C_1}{\arg\min} \left\{ f(\pi, w^k) + \rho D_h(\pi, w^k) \right\},$$
$$w^{k+1} = \underset{w \in C_2}{\arg\min} \left\{ f(\pi^{k+1}, w) + \rho D_h(w, \pi^{k+1}) \right\}. \tag{4}$$

The choice of relative entropy as $h$ also brings the advantage of efficient computation of Bregman projection for simplex constraints, such as $C_1$ and $C_2$, as discussed in (Krichene et al., 2015). These observations result in closed-form updates in each iteration of BAPG in (4). We refer to this specific case as KL-BAPG.

---

**KL-BAPG**

$$\pi \leftarrow \pi \odot \exp(D_X \pi D_Y / \rho), \quad \pi \leftarrow \operatorname{diag}(\mu./\pi 1_m)\pi,$$
$$\pi \leftarrow \pi \odot \exp(D_X \pi D_Y / \rho), \quad \pi \leftarrow \pi \operatorname{diag}(\nu./\pi^T 1_n), \tag{5}$$

---

where $\rho$ is the step size and $\odot$ denotes element-wise (Hadamard) matrix multiplication. KL-BAPG has several advantageous properties that make it ideal for medium to large-scale graph learning tasks. Firstly, it is a single-loop algorithm that only requires matrix-vector/matrix-matrix multiplications and element-wise matrix operations, which are highly optimized on GPUs. Secondly, unlike the entropic regularization parameter in eBPG, KL-BAPG is less sensitive to the choice of the step size $\rho$. Thirdly, KL-BAPG only requires one memory operation for a matrix of size $nm$, which is the main bottleneck in large-scale optimal transport problems rather than floating-point computations. (Mai et al., 2021).

Similar to the quadratic penalty method (Nocedal & Wright, 2006), BAPG is an infeasible method that only converges to a critical point of (1) in an asymptotic sense, meaning there will always be an infeasibility gap if $\rho$ is chosen as a constant. Despite this, BAPG is a suitable option for learning tasks that prioritize efficiency and matching accuracy, such as graph alignment and partition. This idea of sacrificing some feasibility for other benefits is further supported by recent studies such as the relaxed version of GW distance proposed in (Vincent-Cuaz et al., 2021a) for graph partitioning. Additionally, Séjourné et al. (2021) introduced a closely related marginal relaxation, but they did not develop an efficient algorithm with a convergence guarantee. That is, we make $\pi = w$ and $F_\rho(\pi, \pi)$ is the objective introduced in (Séjourné et al., 2021). Our experiments in Sec 4.2 and 4.3 demonstrate that KL-BAPG outperforms existing baselines in graph alignment and partitioning tasks.

## 3 THEORETICAL RESULTS

In this section, we present the theoretical results that have been carried out in this paper. This includes the approximation bound of the fixed-point set of BAPG and its convergence analysis. The cornerstone of our analysis is the following regularity condition for the GW problem in equation (1)

**Proposition 3.1** (Luo-Tseng Error Bound Condition for (1))**.** *There exist scalars $\epsilon > 0$ and $\tau > 0$ such that*

$$\operatorname{dist}(\pi, \mathcal{X}) \le \tau \left\| \pi - \operatorname{proj}_{C_1 \cap C_2}(\pi + D_X \pi D_Y) \right\|, \tag{6}$$

*whenever $\left\| \pi - \operatorname{proj}_{C_1 \cap C_2}(\pi + D_X \pi D_Y) \right\| \le \epsilon$, where $\mathcal{X}$ is the critical point set of (2) defined by*

$$\mathcal{X} = \{\pi \in C_1 \cap C_2 : 0 \in \nabla f(\pi) + \mathcal{N}_{C_1}(\pi) + \mathcal{N}_{C_2}(\pi)\} \tag{7}$$

*and $\mathcal{N}_C(\pi)$ denotes the normal cone to $C$ at $\pi$.*

As the GW problem is a nonconvex quadratic program with polytope constraint, we can invoke Theorem 2.3 in (Luo & Tseng, 1992) to conclude that the error bound condition (6) holds on the whole feasible set $C_1 \cap C_2$. Proposition 3.1 extends (6) to the whole space $\mathbb{R}^{n \times m}$. This regularity condition is trying to bound the distance of any coupling matrix to the critical point set of the GW problem by its optimality residual, which is characterized by the difference for one step projected gradient descent. It turns out that this error bound condition plays an important role in quantifying the approximation bound for the fixed points set of BAPG explicitly.

### 3.1 APPROXIMATION BOUND FOR THE FIXED-POINT SET OF BAPG

To start, we present one key lemma that shall be used in studying the approximation bound of BAPG.

**Lemma 3.2.** *Let $C_1$ and $C_2$ be convex polyhedral sets. There exists a constant $M > 0$ such that*

$$\left\| \operatorname{proj}_{C_1}(x) + \operatorname{proj}_{C_2}(y) - 2\operatorname{proj}_{C_1 \cap C_2}\left(\frac{x+y}{2}\right) \right\| \le M \left\| \operatorname{proj}_{C_1}(x) - \operatorname{proj}_{C_2}(y) \right\|, \quad \forall x \in C_1, y \in C_2.$$

The proof idea follows essentially from the observation that the inequality can be regarded as the stability of the optimal solution for a linear-quadratic problem, i.e.,

$$(p(r), q(r)) = \begin{array}{c} \arg\min_{p,q} \frac{1}{2}\|x - p\|^2 + \frac{1}{2}\|y - q\|^2 \\ \text{s.t.} \quad p - q = r, \, p \in C_1, q \in C_2. \end{array}$$

The parameter $r$ is indeed the perturbation quantity. If $r = 0$, we have $p(0) = q(0) = \text{proj}_{C_1 \cap C_2}(\frac{x+y}{2})$; by choosing $r = \text{proj}_{C_1}(x) - \text{proj}_{C_2}(y)$, it is easy to see that $(p(r), q(r)) = (\text{proj}_{C_1}(x), \text{proj}_{C_2}(y))$. Together with Theorem 4.1 in (Zhang & Luo, 2022), the desired result is obtained. All the proof details are given in Appendix.

Equipped with Lemma 3.2 and Proposition 3.1, it is not hard to obtain the approximation result.

**Proposition 3.3** (Approximation Bound of the Fixed-point Set of BAPG). *The point $(\pi^\star, w^\star)$ belongs to the fixed-point set $\mathcal{X}_{BAPG}$ of BAPG if it satisfies*

$$
\begin{aligned}
\nabla f(w^\star) + \rho(\nabla h(\pi^\star) - \nabla h(w^\star)) + p &= 0, \\
\nabla f(\pi^\star) + \rho(\nabla h(w^\star) - \nabla h(\pi^\star)) + q &= 0,
\end{aligned}
\tag{8}
$$

*where $p \in \mathcal{N}_{C_1}(\pi^\star)$ and $q \in \mathcal{N}_{C_2}(w^\star)$. Then, the infeasibility error satisfies $\|\pi^\star - w^\star\| \leq \frac{\tau_1}{\rho}$ and the gap between $\mathcal{X}_{BAPG}$ and $\mathcal{X}$ satisfies*

$$
\text{dist}\left(\frac{\pi^\star + w^\star}{2}, \mathcal{X}\right) \leq \frac{\tau_2}{\rho},
$$

*where $\tau_1$ and $\tau_2$ are two constants.*

**Remark 3.4.** *If $\pi^\star = w^\star$, then $\mathcal{X}_{BAPG}$ ((8)) is identical to $\mathcal{X}$ and BAPG can reach a critical point of the GW problem (1). Proposition 3.3 indicates that as $\rho \to +\infty$, the infeasibility error term $\|\pi^\star - w^\star\|$ shrinks to zero and thus BAPG converges to a critical point of (1) in an asymptotic way. Furthermore, it explicitly quantifies the approximation gap when we select the parameter $\rho$ as a constant. The proof can be found in Appendix. The explicit form of $\tau_1$ and $\tau_2$ only depend on the problem itself, including $\sigma_{\max}(D_X)\sigma_{\max}(D_Y)$, the constant for the Luo-Tseng error bound condition in Proposition 3.1 and so on.*

### 3.2 Convergence Analysis of BAPG

A natural follow-up question is whether BAPG converges. We answer affirmatively. Under several standard assumptions, we demonstrate that any limit point of BAPG is an element of $\mathcal{X}_{BAPG}$. With this goal in mind, we first establish the sufficient decrease property of the potential function $F_\rho(\cdot)$,

**Proposition 3.5.** *Let $\{(\pi^k, w^k)\}_{k \geq 0}$ be the sequence generated by BAPG. Suppose that $\sum_{k=0}^{\infty} \left(D_h(\pi^{k+1}, w^k) - D_h(w^k, \pi^{k+1})\right)$ is bounded. Then, we have*

$$
F_\rho(\pi^{k+1}, w^{k+1}) - F_\rho(\pi^k, w^k) \leq -\rho D_h(\pi^k, \pi^{k+1}) - \rho D_h(w^k, w^{k+1}).
\tag{9}
$$

As $F_\rho(\cdot)$ is coercive, we have $\sum_{k=0}^{\infty} D_h(\pi^k, \pi^{k+1}) + D_h(w^k, w^{k+1}) < +\infty$. Both $\{D_h(\pi^k, \pi^{k+1})\}_{k \geq 0}$ and $\{D_h(w^k, w^{k+1})\}_{k \geq 0}$ converge to zero. Thus, the following convergence result holds.

**Theorem 3.6** (Subsequence Convergence of BAPG). *Any limit point of the sequence $\{(\pi^k, w^k)\}_{k \geq 0}$ generated by BAPG belongs to $\mathcal{X}_{BAPG}$.*

**Remark 3.7.** *Verifying the boundedness of the accumulative asymmetric error is a challenging task, except in the case where $h$ is quadratic. To address this we perform empirical verification on a 2D toy example, as described in Sec 4.1. The results of this verification for various step sizes, can be seen in Fig. 3 in Appendix C.1. Additionally, when $h$ is quadratic, we can employ the Kurdyka-Łojasiewicz analysis framework, which was developed in (Attouch et al., 2010; 2013) to prove global convergence.*

To the best of our knowledge, the convergence analysis of alternating projected gradient descent methods has only been given under the convex setting, see (Wang & Bertsekas, 2016; Nedić, 2011) for details. In this paper, by heavily exploiting the error bound condition of the GW problem, we take the first step and provide a new path to conduct the analysis of alternating projected descent method for nonconvex problems, which could be of independent interest.

## 4 Experiment Results

In this section, we provide extensive experiment results to validate the effectiveness of the proposed KL-BAPG on various representative graph learning tasks, including graph alignment (Tang et al., 2023) and graph partition (Li et al., 2021). All simulations are implemented using Python 3.9 on a

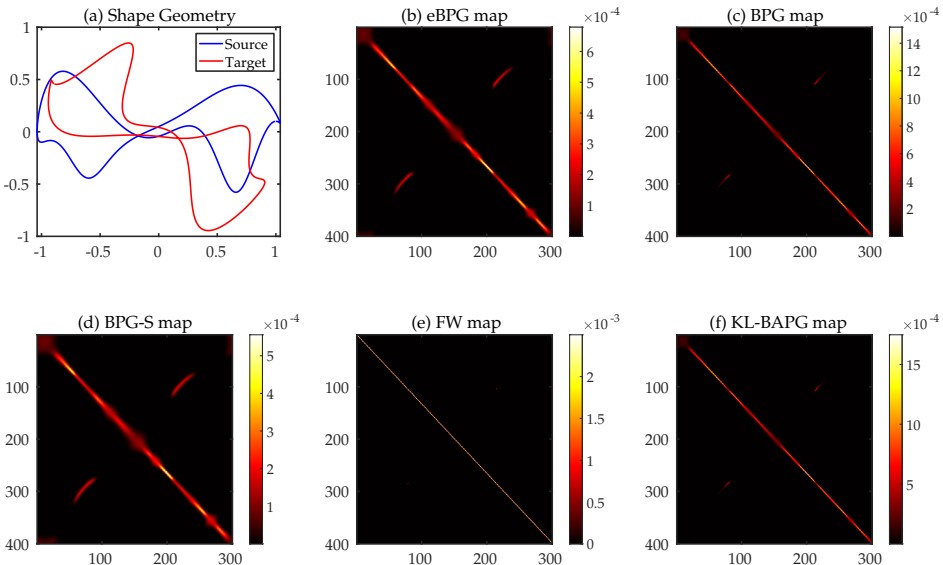

Figure 1: (a): 2D shape geometry of the source and target; (b)-(f): visualization of coupling matrix.

high-performance computing server running Ubuntu 20.04 with an Intel(R) Xeon(R) Gold 6226R CPU and an NVIDIA GeForce RTX 3090 GPU. For all methods conducted in the experiment part, we use the relative error $\|\pi^{k+1} - \pi^k\|_2/\|\pi^k\|_2 \le 1e^{-6}$ and the maximum iteration as the stopping criterion, i.e., $\min\{k \in \mathbb{Z} : \|\pi^{k+1} - \pi^k\|_2/\|\pi^k\|_2 \le 1e^{-6} \text{ and } k \le 2000\}$. Our code is available at https://github.com/squareRoot3/Gromov-Wasserstein-for-Graph.

### 4.1 TOY 2D MATCHING PROBLEM

In this subsection, we study a toy matching problem in 2D to confirm our theoretical insights and results in Sec 2 and 3. Fig. 1 (a) illustrates an example of mapping a two-dimensional shape without any symmetries to a rotated version of the same shape. Here, we sample 300 points from the source shape and 400 points from the target shape, and use the Euclidean distance to construct the distance matrices $D_X$ and $D_Y$.

Figs.1 (b)-(f) provide all color maps of coupling matrices to visualize the matching results. Here, the sparser coupling matrices indicate sharper mapping. All experiment results are consistent with those in Table 4. We can observe that both BPG and FW give us satisfactory solution performance as they aim at solving the GW problem exactly. However, FW and BPG will suffer from a significant computation burden which will be further justified in Sec 4.2 and 4.3. On the other hand, the performance of BPG-S and eBPG is obviously harmed by the inexacness issue. The sharpness of KL-BAPG's coupling matrix is relatively unaffected by its infeasibility issue too much, although its coupling matrix is denser than BPG and FW ones. As we shall see later, the effect of the infeasibility issue is minor when the penalty parameter $\rho$ is not too small and will not even result in any real cost for graph alignment and partition tasks, which only care about the matching accuracy instead of the sharpness of the coupling.

### 4.2 GRAPH ALIGNMENT

Graph alignment aims to identify the node correspondence between two graphs possibly with different topology structures (Zhang et al., 2021; Chen et al., 2020). Instead of solving the restricted quadratic assignment problem (Lawler, 1963; Lacoste-Julien et al., 2006), the GW distance provides an optimal probabilistic correspondence relationship via preservation of the isometric property. Here, we compare the proposed KL-BAPG with all existing baselines: FW (Vayer et al., 2019a), BPG (Xu et al., 2019b), BPG-S (Xu et al., 2019b) (i.e., the only difference between BPG and BPG-S is that the number of inner iterations for BPG-S is just one), ScalaGW (Xu et al., 2019a), SpecGW (Chowdhury & Needham, 2021), and eBPG (Solomon et al., 2016). Except for BPG and eBPG, others are pure heuristic methods without any theoretical guarantee. Besides the GW-based methods, we also

Table 1: Comparison of the matching accuracy (%) and wall-clock time (seconds) on graph alignment. For KL-BAPG, we also report the time of GPU implementation.

| Method | Synthetic | | Proteins | | | Enzymes | | | Reddit | | |
|---|---|---|---|---|---|---|---|---|---|---|---|
| | Acc | Time | Raw | Noisy | Time | Raw | Noisy | Time | Raw | Noisy | Time |
| IPFP | - | - | 43.84 | 29.89 | 87.0 | 40.37 | 27.39 | 23.7 | - | - | - |
| RRWM | - | - | 71.79 | 33.92 | 239.3 | 60.56 | 30.51 | 114.1 | - | - | - |
| SpecMethod | - | - | 72.40 | 22.92 | 40.5 | 71.43 | 21.39 | 9.6 | - | - | - |
| FW | 24.50 | 8182 | 29.96 | 20.24 | 54.2 | 32.17 | 22.80 | 10.8 | 21.51 | 17.17 | 1121 |
| ScalaGW | 17.93 | 12002 | 16.37 | 16.05 | 372.2 | 12.72 | 11.46 | 213.0 | 0.54 | 0.70 | 1109 |
| SpecGW | 13.27 | 1462 | 78.11 | 19.31 | **30.7** | 79.07 | 21.14 | **6.7** | 50.71 | 19.66 | 1074 |
| eBPG | 34.33 | 9502 | 67.48 | 45.85 | 208.2 | 78.25 | 60.46 | 499.7 | 3.76 | 3.34 | 1234 |
| BPG | 57.56 | 22600 | 71.99 | 52.46 | 130.4 | 79.19 | 62.32 | 73.1 | 39.04 | 36.68 | 1907 |
| BPG-S | 61.48 | 18587 | 71.74 | 52.74 | 40.4 | 79.25 | 62.21 | 13.4 | 39.04 | 36.68 | 1431 |
| KL-BAPG | **99.79** | 9024 | **78.18** | **57.16** | 59.1 | **79.66** | **62.85** | 6.9 | **50.93** | **49.45** | 780 |
| KL-BAPG-GPU | - | **1253** | - | - | 75.4 | - | - | 21.8 | - | - | **115** |

consider three widely used non-GW graph alignment baselines, including IPFP (Leordeanu et al., 2009), RRWM (Cho et al., 2010), and SpecMethod (Leordeanu & Hebert, 2005).

**Parameters Setup** We utilize the unweighted symmetric adjacency matrices as our input distance matrices, i.e., $D_X$ and $D_Y$. Alternatively, SpecGW uses the heat kernel $\exp(-L)$ where $L$ is the normalized graph Laplacian matrix. We set both $\mu$ and $\nu$ to be the uniform distribution. For three heuristic methods — BPG-S, ScalaGW, and SpecGW, we follow the same setup reported in their papers. As mentioned, eBPG is very sensitive to the entropic regularization parameter. To get comparable results, we report the best result among the set $\{0.1, 0.01, 0.001\}$ of the regularization parameter. For BPG and KL-BAPG, we use the constant step size 5 and $\rho = 0.1$ respectively. For FW, we use the default implementation in the PythonOT package (see Appendix 4). All the experiment results reported here were the average of 5 independent trials over different random seeds and the standard deviation is collected in Appendix 4.

**Database Statistics** We test all methods on both synthetic and real-world databases. Our setup for the synthetic database is the same as in (Xu et al., 2019b). The source graph $\mathcal{G}_s = \{\mathcal{V}_s, \mathcal{E}_s\}$ is generated by two ideal random models, Gaussian random partition and Barabasi-Albert models, with different scales, i.e., $|\mathcal{V}_s| \in \{500, 1000, 1500, 2000, 2500\}$. Then, we generate the target graph $\mathcal{G}_t = \{\mathcal{V}_t, \mathcal{E}_t\}$ by first adding $q\%$ noisy nodes to the source graph, and then generating $q\%$ noisy edges between the nodes in $\mathcal{V}_t$, i.e., $|\mathcal{V}_t| = (1+q\%)|\mathcal{V}_s|, |\mathcal{E}_t| = (1+q\%)|\mathcal{E}_s|$, where $q \in \{0, 10, 20, 30, 40, 50\}$. For each setup, we generate five synthetic graph pairs over different random seeds. To sum up, the synthetic database contains 300 different graph pairs. We also validate our proposed methods on three other real-world databases from (Chowdhury & Needham, 2021), including two biological graph databases *Proteins* and *Enzymes*, and a social network database *Reddit*. Furthermore, to demonstrate the robustness of our method regarding the noise level, we follow the noise-generating process (i.e., $q = 10\%$) conducted for the synthesis case to create new databases on top of the three real-world databases. Toward that end, the statistics of all databases used for the graph alignment task have been summarized in Appendix 4. We match each node in $\mathcal{G}_s$ with the most likely node in $\mathcal{G}_t$ according to the optimized $\pi^\star$. Given the predicted correspondence set $\mathcal{S}_{\text{pred}}$ and the ground-truth correspondence set $\mathcal{S}_{\text{gt}}$, we calculate the matching accuracy by Acc $= |\mathcal{S}_{\text{gt}} \cap \mathcal{S}_{\text{pred}}|/|\mathcal{S}_{\text{gt}}| \times 100\%$.

**Results of Our Methods** Table 1 shows the comparison of matching accuracy and wall-clock time on four databases. We observe that KL-BAPG works exceptionally well both in terms of computational time and accuracy, especially for two large-scale noisy graph databases *Synthetic* and *Reddit*. Notably, KL-BAPG is robust enough so that it is not necessary to perform parameter tuning. As we mentioned in Sec 2, the effectiveness of GPU acceleration for KL-BAPG is also well corroborated on *Synthetic* and *Reddit*. GPU cannot further speed up the training time of *Proteins* and *Reddit* as graphs in these two databases are small-scale. Additional experiment results to demonstrate the robustness of KL-BAPG and its GPU acceleration will be given in Sec 4.4.

**Comparison with Other Methods** Traditional non-GW graph alignment methods (IPFP, RRWM, and SpecMethod) have the out-of-memory issue on graphs with more than 500 nodes (e.g., Synthetic and Reddit) and are sensitive to noise. The performance of eBPG and ScalaGW is influenced by the

Table 2: Comparison of AMI scores on graph partition datasets using the adjacency matrices and the heat kernel matrices.

| Category | Method | Wikipedia Raw | Wikipedia Noisy | EU-email Raw | EU-email Noisy | Amazon Raw | Amazon Noisy | Village Raw | Village Noisy |
|---|---|---|---|---|---|---|---|---|---|
| Non-GW | FastGreedy | 0.382 | 0.341 | 0.312 | 0.251 | 0.637 | 0.573 | **0.881** | 0.778 |
| | Louvain | 0.377 | 0.329 | 0.447 | 0.382 | 0.622 | **0.584** | **0.881** | **0.827** |
| | Infomap | 0.332 | 0.329 | 0.374 | 0.379 | **0.940** | 0.463 | **0.881** | 0.190 |
| Adjacency | FW | 0.341 | 0.323 | 0.440 | 0.409 | 0.374 | 0.338 | 0.684 | 0.539 |
| | eBPG | 0.461 | **0.413** | **0.517** | 0.422 | 0.429 | **0.387** | 0.703 | 0.658 |
| | BPG | 0.367 | 0.333 | 0.478 | 0.414 | 0.412 | 0.368 | 0.642 | 0.575 |
| | BPG-S | 0.357 | 0.285 | 0.451 | 0.404 | 0.443 | 0.352 | 0.606 | 0.560 |
| | KL-BAPG | **0.469** | 0.396 | 0.508 | **0.428** | **0.457** | 0.362 | **0.736** | **0.681** |
| Heat Kernel | SpecGW | 0.442 | **0.395** | 0.487 | 0.425 | 0.565 | 0.487 | 0.758 | 0.707 |
| | eBPG | 0.100 | 0.082 | 0.011 | 0.188 | 0.604 | 0.031 | 0.002 | 0.003 |
| | BPG | 0.418 | 0.373 | 0.473 | 0.253 | 0.492 | 0.436 | 0.705 | 0.619 |
| | BPG-S | 0.411 | 0.373 | 0.475 | 0.253 | 0.483 | 0.425 | 0.642 | 0.619 |
| | KL-BAPG | **0.533** | 0.365 | **0.533** | **0.436** | **0.630** | **0.502** | **0.797** | **0.711** |

entropic regularization parameter and approximation error respectively, which accounts for their poor performance. Moreover, it is easy to observe that SpecGW works pretty well on the small dataset but the performance degrades dramatically on the large one, e.g., *synthetic*. The reason is that SpecGW relies on a linear programming solver as its subroutine, which is not well-suited for large-scale settings. Besides, although ScalaGW has the lowest per-iteration computational complexity, the recursive $K$-partition mechanism developed in (Xu et al., 2019a) is not friendly to parallel computing. Therefore, ScalaGW does not demonstrate attractive performance on multi-core processors.

## 4.3 GRAPH PARTITION

The GW distance can also be potentially applied to the graph partition task. That is, we are trying to match the source graph with a disconnected target graph having $K$ isolated and self-connected super nodes, where $K$ is the number of clusters (Abrishami et al., 2020; Li et al., 2020). Similarly, we compare the proposed KL-BAPG with the other baselines described in Sec 4.2 on four real-world graph partitioning datasets. Following (Chowdhury & Needham, 2021), we also add three non-GW methods specialized in graph alignment, including FastGreedy (Clauset et al., 2004), Louvain (Blondel et al., 2008), and Infomap (Rosvall & Bergstrom, 2008).

**Parameters Setup** For the input distance matrices $D_X$ and $D_Y$, we test our methods on both the adjacency matrices and the heat kernel matrices proposed in (Chowdhury & Needham, 2021). For KL-BAPG, we pick the lowest converged function value among $\rho \in \{0.1, 0.05, 0.01\}$ for adjacency matrices and $\rho \in \{0.001, 0.0005, 0.0001\}$ for heat kernel matrices. The quality of graph partition results is quantified by computing the adjusted mutual information (AMI) score (Vinh et al., 2010) against the ground-truth partition.

**Results of All Methods** Table 2 shows the comparison of AMI scores among all methods for graph partition. KL-BAPG outperforms other GW-based methods in most cases and is more robust under noisy conditions. Specifically, KL-BAPG is consistently better than both FW and SpecGW, which rely on the Frank-Wolfe method to solve the problem. eBPG has comparable results when using the adjacency matrices, but is sensitive to process the spectral matrices. The possible reason is that the adjacency matrix and the heat kernel matrix have quite different structures, e.g., the former is sparse while the latter is dense. BPG and BPG-S enjoy similar performances in most cases, but they are not as good as our proposed KL-BAPG on all datasets. KL-BAPG also shows competitive performance compared to specialized non-GW graph partition methods. For example, KL-BAPG outperforms Infomap and Louvain in 6 and 4 datasets out of 8, respectively

## 4.4 THE EFFECTIVENESS AND ROBUSTNESS OF KL-BAPG

At first, we target at demonstrating the robustness of KL-BAPG on graph alignment, as it is more reasonable to test the robustness of a method on a database (e.g., graph alignment) rather than a single point (e.g., graph partition).

Table 3: GPU & CPU wall-clock time comparison of KL-BAPG, BPG, and eBPG on graph alignment.

| | Reddit Dataset | | | Synthetic Dataset | | |
|---|---|---|---|---|---|---|
| | KL-BAPG | BPG | eBPG | KL-BAPG | BPG | eBPG |
| CPU Time(s) | 780 | 1907 | 1234 | 9024 | 22600 | 9502 |
| GPU Time(s) | 115 | 1013 | 2274 | 1253 | 4458 | 2709 |
| Acceleration Ratio | **6.78** | 1.88 | 0.54 | **7.20** | 5.07 | 3.51 |

Figure 2: (a) Sensitivity of the noise level and graph scale on the synthetic graph alignment database. (b) Visualization of trade-off among efficiency, accuracy and feasibility on the Reddit database. The infeasibility error is computed by $\|\pi^T 1_n - \nu\| + \|\pi 1_m - \mu\|$. "Closer to the boundary of the outer cycle generally indicates higher accuracy, faster speed, and lower infeasibility error.

**Noise Level and Graph Scale** At the beginning, we present the sensitivity analysis of KL-BAPG with respect to the noise level $q\%$ and the graph scale $|\mathcal{V}_s|$ in Fig. 2 using the Synthetic Database. Surprisingly, the solution performance of KL-BAPG is robust to both the noise level and graph scale. In contrast, the accuracy of other methods degrades dramatically as the noise level or the graph scale increases.

**Trade-off among Efficiency, Accuracy and Feasibility** We present a unified perspective on the trade-off between efficiency, accuracy, and feasibility for all GW-based algorithms on the Reddit database in Fig. 2 (b). As shown, our proposed KL-BAPG is able to achieve a desirable balance between these three factors. Table 7 provides a detailed comparison of the four databases, while Table 8 demonstrates the robustness of KL-BAPG with respect to the step size $\rho$. This experiment supports the validity of Proposition 3.3 and provides practical guidance on choosing the optimal step size. Note that a larger $\rho$ leads to a lower infeasibility error but a slower convergence rate.

**GPU Acceleration of KL-BAPG.** We conduct experiment results to further justify that KL-BAPG is GPU-friendly. In Table 3, We compare the acceleration ratio (i.e., CPU wall-clock time divided by GPU wall-clock time) of KL-BAPG, eBPG, and BPG on two large-scale graph alignment datasets using the same computing server. For eBPG, we use the official implementation in the PythonOT package, which supports running on GPU. For BPG, we implement the GPU version by ourselves using Pytorch. We can find that KL-BAPG has a much higher acceleration ratio on the GPU compared to BPG and eBPG.

## 5 CLOSING REMARK

In this study, we have explored the development of a single-loop algorithm for the relaxation of GW distance computation. By utilizing an error bound condition that was not previously investigated in the GW literature, we have successfully conducted the convergence analysis of BAPG. However, the proposed algorithm still faces the challenge of cubic per-iteration computational complexity, which limits its applicability to large-scale real-world problems. A potential future direction is to incorporate sparse and low-rank structures in the matching matrix to reduce the per-iteration cost and improve the performance of the algorithm. Additionally, our method can also be applied to non-symmetric distance matrices, as the Luo-tseng error bound condition remains valid. On another note, our work also provides a new perspective for the study of the alternating projected descent method for general non-convex problems, which is an open area of research in the optimization community.

## ACKNOWLEDGEMENT

Material in this paper is based upon work supported by the Air Force Office of Scientific Research under award number FA9550-20-1-0397. Additional support is gratefully acknowledged from NSF grants 1915967 and 2118199. Jianheng Tang and Jia Li are supported by NSFC Grant 62206067 and Guangzhou-HKUST(GZ) Joint Funding Scheme. Anthony Man-Cho So is supported by in part by the Hong Kong RGC GRF project CUHK 14203920.

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
