# OpenReview forum: "A Convergent Single-Loop Algorithm for Relaxation of Gromov-Wasserstein in Graph Data "
_ICLR.cc/2023/Conference — ICLR 2023 poster_

### Official Review · Reviewer_Jipi · 2022-10-23

**Confidence:** 2
**Correctness:** 4
**Technical Novelty And Significance:** 3
**Empirical Novelty And Significance:** Not applicable
**Recommendation:** 8

**Clarity, Quality, Novelty And Reproducibility:**

This paper is well-written and structured, with the motivation, as well as the comparison with the existing methods being clearly stated and referenced.

**Strength And Weaknesses:**

The main strength of this work is the splitting strategy for designing the algorithm and the analysis of the algorithm. The Luo-Tseng error bound is utilized to establish the distance between the fixed point of the algorithm and the critical set of the original GW problem. A few minor things in terms of technicality: (1) the algorithm itself is not stated with splitting, e.g., a single variable $\pi$ for the update; but the analysis uses two variables. Can the connection be explicitly stated in the main text? (2) Since the BAPG only returns a solution that is close to the critical set of the GW. Could it happen that some critical point has much worse value than the global min of the GW problem? (3) BAPG returns an infeasible solution in many cases. What if we want to return a feasible solution?

**Summary Of The Paper:**

In this work, the authors propose an efficient single-loop iterative procedure to compute the Gromov Wasserstein (GW) distance between two sets. The GW distance can be formulated as a quadratic programming with a doubly-stochastic constraint. Many of the existing approaches for solving this optimization are double loop in nature, which involve solving another expensive sub-problem iteratively within a single iteration. In the proposed algorithm BAPG, a Bregman alternating projection step is performed with respect to the row and column stochastic constraints in each iteration. By alternatively projecting to polytopes with only row or column constraints, the projection attains a closed form update, thus attaining computational efficiency compared with other algorithms. For a theoretical understanding of the proposed algorithm BAPG, it has been shown that (1) BAPG converges to a fixed point asymptotically and (2) the distance between a fixed point of BAPG and the set of critical points of the original GW problem can be bounded, which is inversely proportional to the penalty parameter. The authors evaluate the algorithm on both synthetic and real datasets related to graph alignment and partition, and demonstrate the solution quality as well as the time efficiency.

**Summary Of The Review:**

This work improves the state-of-the-art for computing the GW distance by obtaining the first provable and efficient single loop algorithm. The analysis of the algorithm is based on the splitting strategy and the Luo-Tseng error bound. With the extensive experiments on both synthetic and real datasets, some of the theoretical claims are verified. It is seen that the proposed method attains good accuracy within less time for graph alignment and partition.

---

> ### Author Response · Authors · 2022-11-15
> **Response to Reviewer Jipi**
>
> Thank you for your positive comments and appreciation of our work! In what follows, we answer the following comment pointed out in your review. In our updated version, we will address all of them.
>
> **Q1**:  *(1) the algorithm itself is not stated with splitting, e.g., a single variable pi
>  for the update; but the analysis uses two variables. Can the connection be explicitly stated in the main text?*
>
> **Response**: Thanks so much for your suggestions. Here is the explicit connection: (5) can be rewritten as one variable update as below,
> $$
> \pi^{k+1} = \mathop{\arg\min}_{\pi \in C_1}  f(\pi,\pi^k) + \rho D_h(\pi, \pi^k), $$
>
> $$
> \pi^{k+1}  = \mathop{\arg\min}_{\pi\in C_2} f(\pi^{k+1},\pi) + \rho D_h(\pi,\pi^{k+1}).
> $$
>
> **Q2**:  *Since the BAPG only returns a solution that is close to the critical set of the GW. Could it happen that some critical point has much worse value than the global min of the GW problem?*
>
> **Response**: Yes, we can always construct a bad initialize point to make the gap large theoretically. However, in practice, it is hard to find an arbitrary feasible point (which means it satisfies the Birkhoff polytope). Instead, all methods started from the same point, i.e.,
> $\pi_0=\frac{1}{nm}\mathbb{I}_{n \times m}$. At least, based on our empirical observation in graph alignment and partition tasks,  the global landscape is not the major concern due to the problem-specific structure.
>
> **Q3**:  *BAPG returns an infeasible solution in many cases. What if we want to return a feasible solution?*
>
> **Response**: Thanks for your instructive comments. Please refer to our response to Q2 for *reviewer h6Kq* and his/her suggestions.  A rounding scheme /Euclidean projection [1]/ Bregman projection (e.g., sinkhorn algorithm) can be invoked here.
>
> [1] Li X, Sun D, Toh K C. On the efficient computation of a generalized Jacobian of the projector over the Birkhoff polytope[J]. Mathematical Programming, 2020, 179(1): 419-446.

---

### Official Review · Reviewer_huho · 2022-10-23

**Confidence:** 2
**Correctness:** 4
**Technical Novelty And Significance:** 4
**Empirical Novelty And Significance:** 4
**Recommendation:** 8

**Clarity, Quality, Novelty And Reproducibility:**


Clarity and Quality.
The paper is well written and attempt is made to make it accessible to a more general audience than experts in numerical computation of graph distances.  The topic discussed is relevant to data science and by extension to ML community.

Novelty.
If the main claim of this paper is correct -- the first theory-based single loop computation of GW distance -- then indeed the paper is quite novel and very useful.

Reproducibility.
I do not know how to measure reproducibility.

**Strength And Weaknesses:**


Strength

o The paper is narrowly focused and laser sharp on what it aims to show

o GW distance on graphs is indeed a useful tool for analysis of large data

o There is a reasonable description of a highly specialized field for a non-expert to follow the main logic of the submission

Weakness

o The paper is highly specialized

o I'm not aware that anyone had previously complained that GW distance computation is a bottleneck is some key data science field

**Summary Of The Paper:**

The paper proposes a new method for computation of the Gromov-Wasserstein distance over graphs.  The GW distance is a measure distance between two distributions defined over different metric spaces. The authors give a list of existing algorithms for this problem which are either expensive computationally or do not have theoretical justification.  The present contribution is stated to be both, thus a breakthrough.  Computational results are provided for a set of different graph problems (alignment, partitioning, etc) which show the proposed scheme is indeed more efficient than existing methods.  The key idea is a surprising application of anold result (error bound condition of Luo & Tseng, 1992).

**Summary Of The Review:**

This is a very technical paper dealing with computation of the GW distance for graph data.  There is a strong theory component to the approach proposed and considerable computational work.  I did not see how the main theorem applied to the algorithm to be single loop but this could be due to my lack of expertise in this area.

---

> ### Author Response · Authors · 2022-11-15
> **Response to Reviewer huho**
>
> We appreciate and thank the reviewer for their positive comments and appreciate both our theoretical and empirical contributions.  We now provide responses to concern you have raised.
>
>  **Q1**:  *I'm not aware that anyone had previously complained that GW distance computation is a bottleneck is some key data science field.*
>
> **Response**: Thanks for your comments. Although previous works did not explicitly state that the GW distance computation is a bottleneck, we would like to emphasize that the GW distance actually plays a crucial role for important graph learning tasks.  First of all, please refer to our response to **Q5** asked by **Reviewer SorA** for details. We add one more real application (multi-omics single-cell integration)  to demonstrate that the main bottleneck of this application is the GW distance computation and a better GW solver can greatly facilitate related downsteam applications. Second,
> we can also conclude from our extensive experiments that a theoretically rigorous and practically efficient algorithm will result in better modeling performance (matching accuracy). Lastly, the most popular algorithm for GW is BPG-S, which is heuristic and lacks a theoretical guarantee. In some sense, our paper filled this theoretical and practical gap.

---

> > ### Comment · Reviewer_huho · 2022-11-21
> > **Acknowledgment**
> >
> > I acknowledge and thank the authors for their clarifying comments.

---

### Official Review · Reviewer_h6Kq · 2022-10-24

**Confidence:** 4
**Correctness:** 4
**Technical Novelty And Significance:** 3
**Empirical Novelty And Significance:** 3
**Recommendation:** 8

**Clarity, Quality, Novelty And Reproducibility:**

# Clarity

Well written work, quite clear overall.

# Quality

From my understanding, this is a good paper that introduce a possibly impactful algorithm supported by relevant theory and appropriate experiments.

# Novelty

To some extend, one could argue that BAPG is quite similar to standard techniques (in particular the Iterative Bregman Iterations interpretation of the Sinkhorn algorithm used in regular computational OT), but the proposed algorithm remains new to the best of my knowledge.

# Reproducibility

Proofs have been (non-extensively) checked and no major flaw was identified (they are pretty well-written). Code to reproduce experiments has been provided with the paper (not tested) and it seems well organized at first sight.

**Strength And Weaknesses:**

## Strength

- Introduces an interesting algorithm to tackle an important problem in Computational OT and related fields.
- Address both theoretical and numerical aspects of the problem.
- Well-written paper.

## Weaknesses

- There are still few theoretical points that remain to be investigated/detailed. For instance:
   - does Eq. (9) enables to get a convergence rate toward the fixed-point $(\pi^\star, w^\star)$? (if so, and provided a rounding scheme is valid there (see below), I think this would yield an overall approximation-rate for GW which may be of interest and could be compared with other benchmarks).
    - Why is it clear that a limit point $(\pi^\infty, w^\infty)$ exists? I guess this follow from some compactness arguments (i.e. the sequence shall be bounded thanks to $f$ being coercive), but this is not striking in the proof as far as I can tell.
- [More a suggestion than a weakness] Experimental evaluation could be slightly extended as they suggest some interesting behavior. In particular, in the light of Fig 2. (b), it seems there is a (natural) tradeoff between convergence speed and the infeasibility error wrt the parameter $\rho$, so a possibly useful idea would be to start with a fairly high $\rho$ and progressively reduce it (as a sort of warm start).
- I think some references are missing, in particular the work of _The unbalanced Gromov Wasserstein distance: Conic formulation and relaxation_ by Séjourné et al., 2021, which introduces a similar relaxation of the GW problem, and proves under specific condition that minimizers of $(\pi, w) \mapsto F(\pi,w)$ are the same as those of $\pi \mapsto F(\pi,\pi)$.

## Other remarks/questions/suggestion
- If my understanding is correct, the output of BAPG is (approximately) a plan of the form $(\pi^\star + w^\star)/2$, which, if I am correct, does not exactly have the expected marginals $(\mu,\nu)$. Assuming this would be critical, could someone apply a `rounding` scheme in the vein of [Altschuler et al., 2018] ?
- [suggestion for clarity] I think that in propositions statements, it may serve clarity to add quantifers "$\forall x,y$", or "$\forall \pi \in \mathbb{R}^{n \times m}$, and so on, to make clear on which parameter does the constant $M,\tau_1,\tau_2$ depend (or not).
- [typo] In Remark 3.7, the ref to prop 3.4 should be 3.5 I think.
- [typo] In proof of prop 3.1, $D_x$ should be $D_X$.

**Summary Of The Paper:**

This article proposes a new algorithm (BAPG) to estimate the GW distance between two discrete measured metric spaces $(X,d_X,\mu)$ and $(Y, d_Y, \nu)$. That is, it consider the following optimization problem  :
$$\min_{\pi} -Tr(D_X \pi D_Y \pi^T)$$
where $D_X = (d_X(x_i,x_j))_{ij}$ (and similarly for $D_Y$) and $\pi$ is constraint to have $\mu,\nu$ as marginals.

The main idea is to consider a relaxation of the bilinear problem that naturally arise from this _quadratic_ optimization problem by minimizing instead

$$F_\rho (\pi,w) = - Tr(D_X \pi D_Y w^T) + \rho D_h(\pi, w)$$

where $\pi$ is constrained to have $\mu$ as first marginal, $w$ is constrained to have $\nu$ as second marginal, $\rho$ is a regularization parameter and $D_h$ is the Bregman divergence associated to the convex map $h$ (in practice, the entropy, so that $D_h$ is the KL divergence, or the quadratic loss so that $D_h$ is the Euclidean distance).

Akin to the usual derivations of the Sinkhorn algorithm as Iterative Bregman Projections [Benamou et al., 2015], authors propose an iterative algorithm (BAPG) to obtained minimizers of the proposed functional.

Of crucial importance, authors show that :
- [Theorem 3.6] Any accumulation point of the sequence $(\pi^k,w^k)_k$ must belong to a set of fixed-points for BAPG.
- [Proposition 3.3] Any such fixed-point must satisfy $|\pi^\star - w^\star| = O(1/\rho)$, so that in particular in the regime $\rho \to \infty$, one expect to have $\pi^\star = w^\star$, and the middle-point $(\pi^\star + w^\star) / 2$ is close (as a $O(1/\rho)$) to an optimal solution of the original GW problem.

Eventually, authors showcase their approach on a variety of numerical experiments, achieving good performances in terms of computational efficiency (running times) and utility (matching accuracy).

**Summary Of The Review:**

I think this is a good paper that introduce an interesting approach to address a difficult and important problem. It is supported by both theoretical and numerical claims.

---

> ### Author Response · Authors · 2022-11-15
> **Response to Reviewer h6Kq:**
>
> Thank you for your positive comments and appreciation of our work! In what follows, we answer the following comment pointed out in your review. In our updated version, we will address all of them.
>
>
> **Q1**: *does Eq. (9) enable you to get a convergence rate toward the fixed-point? (if so, and provided a rounding scheme is valid there (see below, Assuming this would be critical, could someone apply a rounding scheme in the vein of [Altschuler et al., 2018] ?), I think this would yield an overall approximation-rate for GW which may be of interest and could be compared with other benchmarks).*
>
> **Response**: Thanks for pointing out this interesting direction. Based on (9), we can get the standard convergence rate $\mathcal{O}(1/k)$ to the fixed point set instead of the critical point of the original GW problem. If the infeasibility issue does really matter, we believe a rounding scheme /Euclidean projection [1]/ Bregman projection (e.g., sinkhorn algorithm) can be invoked here. However, we would like to clarify that the approximation ratio in other benchmarks (e.g., semidefinite relaxation) is qualified on the ratio between the global optimal value and the approximation one, which we believe it is different from ours (i.e., converge to the citical point set approximately).
>
> [1] Li X, Sun D, Toh K C. On the efficient computation of a generalized Jacobian of the projector over the Birkhoff polytope[J]. Mathematical Programming, 2020, 179(1): 419-446.
>
> **Q2**: *Why is it clear that a limit point exists?*
>
> **Response**: You are correct. The detailed argument has been included in the appendix: as the potential function $F_\rho(\cdot,\cdot)$ is coercive and $\\{(\pi^{k},w^k)\\}_{k \ge 0}$ is a bounded sequence (the Birkhoff polytope is a compact set), we can know the existence of limit points, and from (9), it is easy to get that the limit point belongs to the fixed point set.
>
> **Q3**: *Experimental evaluation could be slightly extended as they suggest some interesting behavior. In particular, in the light of Fig 2. (b), it seems there is a (natural) tradeoff between convergence speed and the infeasibility error wrt the parameter rho, so a possibly useful idea would be to start with a fairly high rho and progressively reduce it (as a sort of warm start).*
>
> **Response**: Thanks for your suggestions. We believe this warm start (adaptive) strategy will work in practice. However, there are several concerns that led us to choosing a constant step size. First of all, the adaptive strategy will bring additional hyperparameters (speed of decay and prefactor). From a theoretical perspective, we do not know (yet) how fast we should update rho. Second, for fair comparison, all of the other baselines utilized the constant step size.
>
> **Q4**: *I think some references are missing, in particular the work of The unbalanced Gromov Wasserstein distance: Conic formulation and relaxation.*
>
> **Response**: Thanks for pointing it out. We have already added the related discussion in our revision. “Moreover, \cite{sejourne2021unbalanced} also proposed a closely related marginal relaxation. That is, we make $\pi = w$ and $F_\rho(\pi,\pi)$ is the objective introduced in \cite{sejourne2021unbalanced}. Unfortunately, they did not develop any provable algorithms for the unbalanced GW distance.”
>
> Other minor issues and comments have been carefully addressed in our revision.

---

> > ### Comment · Reviewer_h6Kq · 2022-11-15
> > **Thanks**
> >
> > Thank you for taking time to answer my review. I agree with your points.

---

### Official Review · Reviewer_sorA · 2022-10-31

**Confidence:** 3
**Correctness:** 3
**Technical Novelty And Significance:** 3
**Empirical Novelty And Significance:** Not applicable
**Recommendation:** 8

**Clarity, Quality, Novelty And Reproducibility:**

-It is not apparent from the paper why being single-loop is important.
-There is no justification on why the two tasks chosen are suitable for the purpose of comparing the relative strength of GW algorithms
-For the two tasks chosen, they are compared only with other GW-based methods. The author(s) should also include performance of other non GW-based methods as benchmark, especially those methods specialized in graph alignment/partitioning (e.g., Infomap for graph-partitioning)
-It is not clear whether the proposed method works well for tasks beyond the two specific graph-based tasks.
-Readers must refer to Xu 2018b to get a clear sense of how the noisy data is generated (not obvious from the contents of this paper)
-Naming conventions are inconsistent with standard literature. For example
   --The paper uses “adjacent matrices” instead of “adjacency matrices”;
   --Using the step size rho as defined in the paper, asymptotic convergence is achieved when rho goes to infinity. Perhaps redefine step size to 1 / rho?
-The text contains several awkwardly phrased sentences and grammatical errors
   --“Arguably, one of the main drawbacks of the double-loop scheme is the computational burden, and not GPU friendly”. (pg. 3 )
   --“The performance of BAPG on the accuracy of mapping falls between {FW, BPG} and {eBPG, BPG-s} but the computational cost is sometimes comparable with BPG-S and FW, even faster in later graph alignment and partition tasks due to its GPU friendly property.” (pg. 6)
- There are some typos in the bibliography (e.g., Vayer Titouan -> Titouan Vayer, Lawrence Carin Duke -> Lawrence Carin)


**Strength And Weaknesses:**

Strengths:
-Technical details are easy to follow and verify
-BAPG involves only the matrix-vector or matrix-matrix operations at each iteration. Consequently, it has good properties such as being GPU implementation-friendly, robustness with respect to the step size, and low memory cost.
-The paper seems to be the first to incorporate the idea of Luo-Tseng error bound condition in the GW literature.
-While the idea of the algorithm is based on applying various known existing tricks and methods, the innovation lies in connecting them in novel ways. In particular, the authors decouple the Birkhoff polytope as simplex constraints for rows and columns separately, which is the key to enabling the projected gradient descent in an alternating fashion.

Weaknesses:
-One of the main claims of the paper is that their algorithm is provably single-loop, as compared to all-but-one algorithms in Table 1 which are classified as double-loop. However, we believe this single-loop vs double-loop terminology is neither standard in the literature nor explicitly defined in the paper. In particular, it wasn’t immediately clear why
 --being single-loop is an important performance measure (either in theory or in practice) that makes it preferable to double-loop;
 --each algorithm in Table 1 is classified as double-loop or single-loop

-Author(s) claim that the infeasibility of the solution provided by their algorithm (BAPG) “does not matter at all” for graph alignment/partitioning, but provide little theoretical justification besides the indifference of these problems to the sharpness of the coupling. Moreover, there does not seem to be sufficient empirical evidence to support their claim. If the infeasibility does not matter at all, does that mean the graph tasks can be potentially solved using an easier formulation (i.e., one that does not require solving an GW problem)? This seems to suggest that the experiments chosen are unsuitable for the purpose of comparing the relative strength of GW algorithms. Also, the authors compared their graph alignment/partitioning performance with other GW-based algorithms. It will be much more meaningful to include comparisons with other non GW-based methods, especially those methods specialized in graph alignment/partitioning (e.g., Infomap for graph-partitioning), especially if the authors claim their method achieves state-of-the-art result
- Because the author(s) assume only that Dx and Dy are symmetric distance matrices, their algorithm could potentially be extended to a wider class of data/tasks, but they make little mention of this. Maybe include this in future work.
-Author(s) claim that their method is stable compared to existing methods, but provide no such notion of stability
-The accuracy of their algorithm in figure 2a seems a little suspicious


**Summary Of The Paper:**

The paper proposes an (approximation) algorithm to compute Gromov-Wasserstein (GW) distance, and called it Bregman Alternating Projected Gradient (BAPG) method. The author(s) claim that this is the first single-loop algorithm that has provable convergence guarantees. In particular, they provide an approximation bound for the distance between the fixed-point set of BAPG method and the critical point set of GW. Their analysis is based on the observation that the GW problem satisfies a so-called Luo-Tseng error bound condition, and the authors claim that they are the first to incorporate this idea in the GW literature. The paper complements their theoretical analysis with a few experiments comparing the performance of BAPG and other algorithms on graph alignment and partition tasks. They claim that BAPG achieves state-of-the-art results in terms of both wall-clock time and quality of solutions.



**Summary Of The Review:**

The algorithm proposed connects existing optimization ideas in a novel way, and the analysis seems technically sound. It is also expected to run faster than competing GW algorithms in practice. The author(s) claim that this the first single-loop GW algorithm that has provable convergence guarantees, though being single-loop was not a standard performance measure in the literature nor well explained in the paper. Finally, it wasn’t clear that the experiments chosen are suitable for the purpose of comparing the relative strength of GW algorithms.

Score increased after review.

---

> ### Author Response · Authors · 2022-11-15
> **Response to Reviewer sorA (4): a New Task --- Multi-Omics Single-Cell Integration II**
>
> **Experimental Results on Multi-Omics Single-Cell Integration**：
>
>
> Table 2: Comparison of the label transfer accuracy (\%) and wall-clock time (seconds) on the scGEM and scNMT datasets for single-cell multi-omics integration. DNA, Gene, and Chr. represent the modalities of DNA Methylation, Gene Expression, and Chromatin Accessibility, respectively.
>
> | Dataset     | scGEM    |          |         | scMNT    |          |           |           |         |
> |-------------|----------|----------|---------|----------|----------|-----------|-----------|---------|
> | Method      | DNA→Gene | Gene→DNA | Time    | DNA→Gene | Gene→DNA | Chr.→Gene | Gene→Chr. | Time    |
> | FW          | 40.7     | 40.7     | 0.7     | 74.2     | 74.3     | 44.1      | 45.4      | 19.9    |
> | eBPG        | 7.9      | 10.7     | 7.4     | 63.4     | 68       | 46.6      | 44.8      | 64.2    |
> | specBPG     | 21.5     | 21.5     | 0.4     | 61.4     | 57.2     | 36.4      | 39.7      | 2.2     |
> | BPG-S       | 9.0      | 14.1     | 1.2     | 69.0     | 73.2     | 60.1      | 60.7      | 2.3     |
> | BPG         | 11.9     | 11.3     | 11.5    | 75.1     | 73.2     | 60.8      | 60.0      | 22.3    |
> | BAPG        | 52.0     | 50.8     | **0.3** | 74.0     | 74.3     | 68.0      | 67.0      | **0.8** |
> | UnionCom    | 48.0     | 50.8     | 6.4     | 85.6     | 76.3     | 80.1      | 78.8      | 147.4   |
> | Pamona      | 61.6     | **58.2** | 4.4     | 71.6     | 56.6     | 71.9      | 66.8      | 26.1    |
> | Pamona-BAPG | **67.2** | **58.2** | 0.7     | **85.7** | **91.3** | **80.7**  | **82.9**  | 6.5     |
>
> Table 2 summarizes the results of all compared methods on the scGWM and scNMT datasets. Among all GW methods, BAPG achieves the best performance on 4 out of 6 scores and has the shortest running time. UnionCom and Pamona are better than most pure GW methods as they involve additional pre-processing and post-processing techniques specified for this task. Moreover, Pamona-BAPG outperforms Pamona on 5 out of 6 scores and reduces the running time to 16\%~25\%. It indicates that the calculation of  GW distance is a bottleneck for this task, and a better GW solver can greatly facilitate related downstream applications. We also would like to mention that it is just a preliminary study to show the potential of applying BAPG to other tasks, due to the time budget.   We left other potential application-driven tasks as our future work. If the reviewer can provide other GW distance-based learning tasks, we’d really appreciate it.
>
> Other minor issues (typos and citations)  and comments have been carefully addressed in our revision. Please let us know if our response addresses your concerns. We are happy to address any remaining points during the discussion phase. If our response has adequately addressed your concerns, we kindly ask you to consider raising the score.

---

> > ### Comment · Reviewer_sorA · 2022-11-21
> > **Will change the score accordingly.**
> >
> > Thank you for taking the time to answer my review. I appreciate that you made the notion of “single-loop” more explicit in your revision, and I completely agree with the advantage of an algorithm being single-loop.
> >
> > Q2: Thanks for the clarification.
> > Q3: Thanks for considering my suggestions. Overall, I’m pleased that BAPG performs well against Infomap and Louvain. (Do you mean “BAPG outperforms Infomap and Louvain in 6 *and 4* out of 8 datasets, respectively”?
> > Q4: Thank you for clarifying this. It was not immediately clear to me that by stable, you meant robust to noise level and graph scale. Do you have any intuition for why your method is so robust to these parameters while other methods are not?
> > Q5: This is interesting; thanks for adding it. Also, thanks for addressing the minor issues too. All things considered, I am happy to raise the score to 7 as the paper aims to tackle an important OT problem and proposes an innovative and practical algorithm.

---

> > > ### Author Response · Authors · 2022-12-13
> > > **Response to Reviewer sorA**
> > >
> > > Thanks so much for appreciating our work and being willing to increase your score.
> > >
> > > *Q3: Thanks for considering my suggestions. Overall, I’m pleased that BAPG performs well against Infomap and Louvain. (Do you mean “BAPG outperforms Infomap and Louvain in 6 and 4 out of 8 datasets, respectively”?*
> > >
> > > **Response**: Thanks for pointing it out. This is a typo and we will update our submission accordingly.
> > >
> > > *Q4: Thank you for clarifying this. It was not immediately clear to me that by stable, you meant robust to noise level and graph scale. Do you have any intuition for why your method is so robust to these parameters while other methods are not?*
> > >
> > > **Response**: Thanks so much for your suggestion. As we discussed the advantages of the “single-loop” algorithmic scheme, the only hyperparameter for the proposed BAPG is the step size rho. However, other benchmarks will have at least two additional hyperparameters for the inner loop, which partially leads to their degraded performance.

---

> ### Author Response · Authors · 2022-11-15
> **Response to Reviewer sorA (3): a New Task --- Multi-Omics Single-Cell Integration**
>
> **Q5**: *It is not clear whether the proposed method works well for tasks beyond the two specific graph-based tasks.*
>
> **Response**: Thanks so much for your suggestion. We would like to highlight that our main contribution lies in provable single-loop algorithm design for computing the GW distance approximately instead of developing new GW-based models for specific graph-learning tasks. The main point we would like to deliver in this paper is that we can sacrifice the infeasibility a little bit to gain on model performance and computational efficiency for those tasks that do not care about the sharpness of the matching. We give two important graph learning tasks (graph alignment and partition) as examples to validate our theoretical and algorithmic contributions and findings. Moreover, the GW distance arises from a relaxation of classic quadratic assignment problems [1]. Instead of giving back to the permutation matrix (i.e., hard matching result), GW will provide an alternative probabilistic (soft) correspondence. We believe that graph alignment (the matching) is the most relevant task to test the performance of GW-based algorithms, especially given our objective. Moreover, we also follow other GW-based papers in graph learning and add the graph partition task to illustrate further and justify the effectiveness of BAPG.
>
> To address the reviewer's concern, we’d love to further evaluate the proposed BAPG on an additional task (alignment/matching based), **multi-omics single-cell integration**, to validate the application capability of BAPG beyond two specific tasks. It aims at integrating multiple molecular features at different modalities in a cell, e.g., gene expressions and chromatin accessibility [2-5]. This task offers opportunities for gaining holistic views of cells and is selected as ``Method of the Year 2019'' by Nature [2].
>
> **Task description**:
>
> Suppose we have two single cell multi omics datasets, $X = [x_1;…;x_{n_x}] \in \mathbb R^{n_x \times d_x} $ and $Y = [y_1;…;y_{n_y}] \in \mathbb R^{n_x \times d_x}$ across two modalities, where $n_x(n_y)$ and $d_x(d_y)$ are the number of cell samples and feature dimensions for $X(Y)$, respectively. Given $X$ and $Y$, the task of multi-omics single-cell integration aims at calculating the joint representation of two modalities. In the joint representation, the same cell types in $X$ and $Y$ should be clustered together. Thus, aligning the same types of cells across $X$ and $Y$ in advance can improve the quality of the joint representation.
>
> **Dataset and Evaluation**:
>
> Table 1: Statistics of two real-world single-cell multi-omics datasets.
> |                 | scGEM           |                 | scMNT           |                 |                         |
> |-----------------|-----------------|-----------------|-----------------|-----------------|-------------------------|
> | Modalities Name | Gene Expression | DNA Methylation | Gene Expression | DNA methylation | Chromatin Accessibility |
> | Samples $n$     | 177             | 177             | 612             | 709             | 1940                    |
> | Features $d$    | 34              | 27              | 300             | 300             | 300                     |
>
> Following [4,5], we conduct experiments on two real-world single-cell multi-omics datasets: (1) the **scGEM** dataset for the single-cell analysis of genotype, expression and methylation data and (2) the **scNMT** dataset for the single-cell analysis of nucleosome, methylome and transcriptome data. The dataset statistics is listed in Table 1. Same as [4,5], we use the label transfer accuracy (i.e., the percentage of aligned cell pairs with the same cell type across two modalities) to measure the ability to transfer labels of the shared cells from one modality to another.
>
> **Baselines**:
>
> We compare the proposed BAPG with other GW methods in our paper by directly setting the distance matrix $D_X$ and $D_Y$ as the feature similarity matrix. Besides, we add two baselines specified for this task, including UnionCom[4] and Pamona[5]. Pamona also relies on calculating the GW distance method and uses eBPG as the GW solver. We replace eBPG in Panona with BAPG and obtain a new method named Pamona-BAPG.
>
>
> [1] Foundations of computational mathematics, 2011, 11(4): 417-487.
>
> [2] Wen H, Ding J, Jin W, et al. Graph neural networks for multimodal single-cell data integration[C]//Proceedings of the 28th ACM SIGKDD Conference on Knowledge Discovery and Data Mining. 2022: 4153-4163.
>
> [3] Teichmann S, Efremova M. Method of the Year 2019: single-cell multimodal omics[J]. Nature Methods, 2020.
>
> [4] Cao K, Bai X, Hong Y, et al. Unsupervised topological alignment for single-cell multi-omics integration[J]. Bioinformatics, 2020.
>
> [5] Cao K, Hong Y, Wan L. Manifold alignment for heterogeneous single-cell multi-omics data integration using Pamona[J]. Bioinformatics, 2022, 38(1): 211-219.

---

> ### Author Response · Authors · 2022-11-15
> **Response to Reviewer sorA (2)**
>
> **Q2**: *Author(s) claim that the infeasibility of the solution provided by their algorithm (BAPG) “does not matter at all” for graph alignment/partitioning, but provide little theoretical justification besides the indifference of these problems to the sharpness of the coupling. Moreover, there does not seem to be sufficient empirical evidence to support their claim. If the infeasibility does not matter at all, does that mean the graph tasks can be potentially solved using an easier formulation (i.e., one that does not require solving a GW problem)? This seems to suggest that the experiments chosen are unsuitable for the purpose of comparing the relative strength of GW algorithms.*
>
> **Response**: First of all, we would like to apologize for our poor choice of words “*does not matter at all*”. We correct it in the manuscript. Let us elaborate on what we meant. The effect of the infeasibility of the BAPG solution is minor when the penalty parameter rho is not too small. Please see Table 4 for detailed experimental results. It can also be observed that when $\rho$ is too small (e.g., $\rho$ = 0.01), the matching accuracy will dramatically decrease. However, when $\rho$ is larger than a certain threshold, the model performance is not sensitive with respect to the infeasibility error or $\rho$. This is a feature, which, we believe can be used to calibrate the choice of $\rho$. In short, our paper proposed BAPG, which is able to sacrifice some feasibility (not too much) to gain both matching accuracy and computational efficiency for those tasks that only care about the matching accuracy (i.e. the performance measure) but not the sharpness of the matching coupling. While this trade-off is discussed in the Introduction, we will refer the reader to this discussion in the empirical section. For these practically relevant tasks (graph alignment and partition), we have already conducted and reported extensive experiments to demonstrate BAPG is able to obtain the desired performance and achieve a great balance between accuracy and efficiency. Finally, the GW distance can be regarded as a relaxation of classic quadratic assignment problems. Instead of giving back to the permutation matrix (hard matching result), GW will provide an alternative probabilistic correspondence. We believe that graph alignment is the most relevant task to test the performance of GW-based algorithms. Moreover, we are already also following the types of experiments performed in the GW literature on graph learning and added the graph partition task to illustrate further and justify the effectiveness of BAPG.
>
>
> **Q3**:  *Also, the authors compared their graph alignment/partitioning performance with other GW-based algorithms. It will be much more meaningful to include comparisons with other non-GW-based methods, especially those methods specialized in graph alignment/partitioning (e.g., Infomap for graph-partitioning), especially if the authors claim their method achieves state-of-the-art results.*
>
> **Response**: Thanks so much for your suggestions. Upon your suggestions, we add other non-GW-based baseline methods (e.g., Infomap) for graph partition tasks to further demonstrate the effectiveness of the proposed BAPG, see Table 2 and Table 3 in the revision for details. We can observe that BAPG also shows competitive performance compared to the specialized non-GW graph partition methods. For example, BAPG outperforms Infomap and Louvain in 6 out of 8 scores, respectively. For the graph alignment task, we also add three widely used non-GW-based methods for comparison, and none of them are comparable to BAPG.
>
> **Q4**: *Author(s) claim that their method is stable compared to existing methods, but provide no such notion of stability -The accuracy of their algorithm in figure 2a seems a little suspicious.*
>
> **Response**: Thanks for your comments. We may require further clarification from the reviewer in order to answer this question appropriately. What does the reviewer mean by “provide no such notion of stability”? We believe we give a detailed definition of (matching) accuracy in Section 4.2. In Figure 2(a),  we conduct the sensitivity analysis of BAPG w.r.t the noise level and graph scale. We can observe that the solution performance of BAPG (matching accuracy) is robust (not sensitive) to both the noise level and graph scale. In contrast, the accuracy of other methods degrades dramatically as the noise level or the graph scale increases. We also provide the source code to reproduce all experiment results.

---

> ### Author Response · Authors · 2022-11-15
> **Response to Reviewer sorA (1)**
>
> Thanks for your instructive comments and appreciating our algorithmic contributions. Hopefully the following discussion can clear up your concerns. Also, we modify our submission based on your suggestions and comments in the latest version.
>
> **Q1:**  *One of the main claims of the paper is that their algorithm is provably single-loop, as compared to all-but-one algorithms in Table 1 which are classified as double-loop. However, we believe this single-loop vs double-loop terminology is neither standard in the literature nor explicitly defined in the paper.  In particular, it wasn’t immediately clear why being single-loop is an important performance measure (either in theory or in practice) that makes it preferable to double-loop; --each algorithm in Table 1 is classified as double-loop or single-loop*
>
> **Response:** Thanks for pointing out this issue. We now have  defined more explicitly what is meant by a single-loop algorithm in our revision according to your suggestion – ``Single loop algorithms do not require another iterative algorithm as a subroutine at each iteration’’. Regarding the terminology, we adopt what we believe is a convention in the machine learning and optimization literatures in which the  “ single-loop vs double-loop terminology” appears to be adopted, we list a few papers in the last few years that follow this convention see, for example,[1-6] for more details.
>
> In summary, we think that a single-loop scheme is advantageous because of the following reasons: (1) **Robustness to hyperparameters**; without the inner iterative subroutine (i.e. the inner loop), we can dispense of costly hyperparameter tuning processes associated to the inner loop. The only hyperparameter for the proposed BAPG is the step size rho. However, for any other double-loop algorithms we have to specify (at the very least) the tolerance error, step size in the inner loop, and other other parameters. In short, the single loop algorithm is more robust (in the sense of ease of calibration) than those based on a double loop. (2) **GPU implementation friendly**;  the single loop algorithm is friendlier than double-loop algorithms  to GPU acceleration. In table 5, we can observe that BAPG has a much higher acceleration ratio on the GPU environment compared with BPG and eBPG (two double-loop algorithms). (3) **BPG-S lacks a theoretical guarantee**; BPG-S (single loop based heuristic algorithm)  has already been widely used in practice and achieves a superior performance on a diverse set of GW-based graph learning tasks. However, such a heuristic still lacks theoretical support, and thus it is not necessarily guaranteed to perform well (or even converge) under naturally noisy observations. The proposed BAPG fulfills this theoretical gap and achieves a better performance compared with BPG-S.
>
> Thus, we believe (as also expressed by other reviewers) that the theoretical and algorithmic contributions of the single loop BAPG  are substantial and useful to the (significant and growing) community applying GW-type formulations.
>
> [1] Li J, Gu B, Huang H. A fully single loop algorithm for bilevel optimization without hessian inverse[C]//Proceedings of the AAAI Conference on Artificial Intelligence. 2022, 36(7): 7426-7434.
>
> [2] Yang J, Orvieto A, Lucchi A, et al. Faster single-loop algorithms for minimax optimization without strong concavity[C]//International Conference on Artificial Intelligence and Statistics. PMLR, 2022: 5485-5517.
>
> [3] Xu Z, Zhang H, Xu Y, et al. A unified single-loop alternating gradient projection algorithm for nonconvex-concave and convex-nonconcave minimax problems[J]. arXiv preprint arXiv:2006.02032, 2020.
>
> [4] Tran Dinh Q, Liu D, Nguyen L. Hybrid variance-reduced SGD algorithms for minimax problems with nonconvex-linear function[J]. Advances in Neural Information Processing Systems, 2020, 33: 11096-11107.
>
> [5] Zhang J, Xiao P, Sun R, et al. A single-loop smoothed gradient descent-ascent algorithm for nonconvex-concave min-max problems[J]. Advances in Neural Information Processing Systems, 2020, 33: 7377-7389.
>
> [6]Shen J, Wang Z, Xu Z. Zeroth-order single-loop algorithms for nonconvex-linear minimax problems[J]. Journal of Global Optimization, 2022: 1-30.

---

> ### Author Response · Authors · 2022-11-19
> **Could you please check our response?**
>
> Dear Reviewer,
>
> Since only a few hours remain in the discussion period, we would appreciate it if you check and reply to our response to your comments soon. This will give us time to address further questions and comments that you may have before the end of the discussion period. If our response adequately addresses your concerns, please consider raising the score of our submission. Thank you very much for your time.
>
> Best,
> All the authors.

---

### Decision · Program_Chairs · 2023-01-22

**Decision:**

Accept: poster

**Justification For Why Not Higher Score:**

N/A

**Justification For Why Not Lower Score:**

N/A

**Metareview: Summary, Strengths And Weaknesses:**

This paper proposes a *convergent* single-loop algorithm for Gromov-Wasserstein in Graph Data.

# Summary
The GW problem is important and challenging (non-convex quadratic objective in space of n x n bistochastic matrices).
This paper proposes a provably convergent (to local minimum) single loop approach, which could be of great practical interest, because single-loop algorithms are easier to tune (no need for inner loop's # of iterations, or tolerance). The non-asymptotic convergence theory that is provided is original, but some limitations:
- The authors mention at the end of their introduction, in Table 1 and elsewhere in the paper, that their algorithm does not always return a feasible solution for GW (e.g. a bistochastic matrix).
- The author mention in R.3.7 that the algorithm used in experiments is not provably convergent (theory supports a symmetric Bregman divergence, i.e. L2; their implementation uses KL).
The authors show speed-ups as well as comparable or better performance (in accuracy) to other approaches in graph alignment /partitioning tasks.

# Process

The paper was supported by all reviewers during the rebuttal phase. After the rebuttal phase, AC and reviewers discussed shortcomings. This led to 2 reviewers (h6kQ and Jipi) lowering their grades. Because the system was locked, this is only reflected in messages to the authors ("Unfortunately have to lower my score", "Lower my score due to my misunderstanding"). Reviewer h6kQ communicated to me a new grade of 5. One reviewer participated in a discussion with the others but stopped participating. Another reviewer did not join the discussion after the rebuttal period. Some of the concerns were then discussed with the authors at a later date.

The AC and SAC now acknowledge that these discussions came late in the review process, and therefore did not let enough opportunity to the authors to effectively amend their paper, and address the late concerns.

# Summary of shortcomings discussed (with and without authors)

- The "provable" algorithm featured in {the title, the introduction, Table 1} is Eq.5 uses a sym. Bregman div. (a.k.a. L2). The algorithm used in experiments uses KL projections (Eq. 2). The authors argued during a discussion that these 2 algorithms were similar enough to conflate them, and use a single name ("BAPG"). I disagree. Projections onto the simplex are crucial, here they differ significantly.  While modifying the title could be easily fixed (e.g. "On the convergence of single-loop relaxations of GW for graph data"), significant parts of the paper need to be rewritten to lift that ambiguity.

- The feasibility issue has raised several questions from reviewers and AC. For instance, the only numerical evidence of (in)feasibility was Figure 2b, but it was reported by the authors to be wrong. To balance this, the authors argued that their contribution was to be understood as a relaxation of GW, tailored for graph data. Unfortunately, the paper is not written that way. The introduction mentions GW 13 times, provides shortcomings for existing GW approaches, and follows with *"To bridge the above theoretical-computational gap, we propose the first provable single-loop algorithm for the GW distance computation, which we call ..."*. Yet, a few sentences later, the authors write *"Nonetheless, the iterates generated by BAPG do not necessarily satisfy the Birkhoff polytope constraint"*. This contradiction appears in the paper: most comparisons focus on feasible solvers.

Comments on experiments:
- Hyperparameter search is mentioned in paper, but not in code, where they are are hardcoded. This is unreliable, given the amount of copy pasted code.
- No results for eBPG on "heat kernel" in Table 4. The argument that is mentioned (kernel sparse or dense) is unclear.
- BAPG (PyTorch) is compared to eBPG in POT (rather than reimplemented with PyTorch). The authors write "For eBPG, we use the official implementation in the PythonOTpackage, which supports running on GPU.". POT reads: "This function is backend-compatible and will work on arrays from all compatible backends. But the algorithm uses the C++ CPU backend which can lead to copy overhead on GPU arrays."
- In the "3-D space" of {accuracy /  speed / feasibility} the paper uses a partial view. A unified perspective is lacking. For instance, this could be corrected by adding the infeasibility of all solutions that are reported, along with time/accuracy.

# Decision after revision
After revision, SAC and PC agreed that above concerns are sufficiently adressed and that the paper is ready for acceptance.

**Summary Of Ac-Reviewer Meeting:**

For the most part, I was not able to get reviewers react to my several requests (about 10 emails).